# The Rise of Quantum Information and Communication Technologies

**Antonio Manzalini *** and **Luigi Artusio**

TIM—Telecom Italia, 10148 Turin, Italy
* Correspondence: antonio.manzalini@telecomitalia.it

**Abstract:** Today, we are already using several-component devices and systems based on the technologies developed during the first quantum revolution. Examples include microchips for servers, laptops and smartphones, medical imaging devices, LED, lasers, etc. Now, a second quantum revolution is progressing fast, exploiting technological advances for the ability to engineer and manipulate other quantum phenomena such as superposition, entanglement and measurement. As a matter of fact, there is an impressive increase in research and development activities, innovation, public and private investments in a new wave of quantum services and applications. In this scenario, quantum information and communication technologies (QICTs) can be defined as a set of technological components, devices, systems and methods for elaborating, storing and transmitting/sharing quantum information. This paper addresses the challenges and opportunities enabling the rise of QICTs. In order to provide a concrete example, the paper describes an overview of the European project EQUO (European Quantum ecOsystems) dealing with ongoing innovation activities in the QICT avenue; in fact, EQUO aims at developing and demonstrating the feasibility of QKD (quantum key distribution) networks and their related integration in current telecommunications infrastructures towards the quantum internet.

**Keywords:** quantum information; quantum key distribution; quantum information and communication technologies

## 1. Introduction

Information communication technologies (ICT) and telecommunication industries have been always considered engines' socio-economic growth and transformation. Today, for instance, there is a pervasive diffusion of ultra-broadband network connectivity, an exploitation of the high-bandwidth and low-latency 5G infrastructures of the cloud-edge computing and a wide adoption of artificial intelligence. All of these comprise the so-called digital transformation, a technological revolution expected to produce socio-economic growth.

At the same time, the sustainability of such future network and service scenarios will have to face several challenges, such as the transmission and processing of an increasing quantity of data, the requirements of ultra-low latencies, the automation of management operations, the need for solutions for resilience, security, and privac0y, the optimization of energy consumption, etc.

It should be mentioned that this digital transformation is still based on electronics and the well-known risk of the end of Moore's law. This may jeopardize long-term sustainability. Therefore, rethinking the ways of performing computation, communications and networking has already started, and quantum technologies are expected to mitigate or solve some of above techno-economic challenges, even beyond the end of Moore's law.

Today a second quantum revolution is progressing fast, exploiting the technological advances for the ability to engineer and manipulate other quantum phenomena such as superposition, entanglement and measurement. As a matter of fact, we are witnessing

an impressive increase in research and development activities, innovations efforts, public and private investments in new quantum services and applications (e.g., communications, computing, simulations, sensing and metrology) [1,2].

In this scenario, quantum information and communication technologies (QICTs) can be defined as a set of technological components, devices, systems and methods for elaborating, storing and transmitting/sharing quantum information.

This paper focuses on QICT services and applications for quantum communications, specifically in the domain of quantum security, which appear to be more mature today as they rely on systems such as quantum key distribution (QKD) [3].

In particular, QKD is a secure communication method that implements a cryptographic protocol involving components of quantum mechanics. It enables two parties to produce a shared random secret key known only by them, which then can be used to encrypt and decrypt messages. The sender (traditionally referred to as Alice) and the receiver (referred to as Bob) are connected by a quantum communication channel that allows quantum states to be transmitted and an authenticated classical channel for deriving a common secret from the exchanged quantum information.

QKD systems can be divided into two main classes: discrete-variable and continuous-variable. In the former class, the quantum information is typically encoded into discrete optical modes of a single photon, e.g., polarization or time bin; in this case, single-photon detectors are required for decoding. In the latter class, quantum states are described in an optical domain where the eigenstates are continuous with an infinite dimension, e.g., using Gaussian optical states.

In order to provide another concrete example of QKD applications, the paper describes an overview of the European project EQUO (European Quantum ecOsystems), dealing with ongoing innovation activities in the QICT avenue; in fact, EQUO aims at developing and demonstrating the feasibility of discrete-variable QKD networks and their related integration into current telecommunications infrastructures towards the quantum internet.

In particular, the integration approach is based on the Software-Defined Networking (SDN) paradigm, where network programmability is enabled through logically centralized control, management and orchestration planes. As a matter of fact, this is the trend also adopted for current production telecommunication networks where multiple managers and controllers are deployed for scalability and reliability reasons, who in turn rely on distributed consensus protocols to operate in a logically centralized manner. The EQUO vision also extends this SDN paradigm for the QKD network and will do so in the future for the quantum internet. The vision, the developments and the experimental demonstrations of the project EQUO are expected to bring significant advances in the direction of the industrialization and deployment of the Euro Quantum Communication Infrastructure, essential for the protection of the European digital infrastructure, and the development of an industrial European ecosystem, including a thriving small–medium enterprise sector.

## 2. Challenges and Opportunities Enabling the Rise of QICTs

Society and today's industry are experiencing a profound techno-economic transformation. Technologies such as software-defined networks (SDN) and Network Function Virtualization (NFV) allow the design and deployment of 5G infrastructure and Cloud-edge computing, with high levels of programmability and flexibility when providing any sort of digital service. The need to look at the long-term evolution and sustainability of these ICT and telecommunication scenarios is stimulating a rethinking of the ways of performing computation, communications and networking based on quantum technologies.

There is evidence that second quantum revolution is progressing fast, based on the capabilities to detect and manipulate single quantum particles (e.g., electrons, photons, ions, etc.), not achieved during the second quantum revolution. Specifically, there are three main quantum phenomena that a second revolution aims at engineering and controlling: superposition, entanglement [4] and measurement.

It is expected that, in less than 10 years, we will see a significative impact on many markets, including security, finance, medicine, energy, transportation, etc. International innovation activities and standardization bodies agree on identifying four main application areas: communications, computing, simulations and sensing and metrology.

Quantum communications include two main subdomains: (1) quantum-safe communications and (2) the teleportation of quantum information (e.g., quantum internet) [5]. Quantum-safe communications are leveraging systems such as quantum key distribution (QKD) and quantum random number generators (QRNGs), which have rather high technology readiness levels (TRLs). We should be reminded that TRLs represent a systematic metric/measurement convention for assessments of the maturity of a particular technology and the consistent comparison of maturity between different types of technology [6]. QKD is a symmetric secret key negotiation protocol and technology offering unconditional secrecy based on the laws of quantum mechanics. QKD is an area in which interest and investments are growing very rapidly as shown through recent developments, both theoretically and experimentally. Concerning the state of the art in QKD, reference [7] provides a rigorous threat analysis based on the most recent recommendations and practical network deployment security issues.

For the second application area, quantum computers operate faster than classical computers do when solving complex optimization and combinatorial problems (the processing time is reduced from exponential to polynomial time). Trapped ions and superconducting qubits seem to be quantum technologies which that satisfy the five required criteria for quantum computing, formerly defined by Di Vincenzo [8]; also, other approaches are gaining momentum, such as impurity spins in solids, neutral Rydberg atoms and topological photonics. The application area of quantum simulations regards those applications where well-controlled quantum systems can be used to simulate other systems that are less accessible and more complex for direct simulation. Quantum sensing and metrology concern applications where the high sensitivities of quantum systems are needed to measure physical properties with high precision (e.g., magnetic and heat sensors, gravimeters, clocks, GPS-free navigators, etc.).

Overall, some quantum services and applications are becoming commercially available (e.g., QKD, QRNG, quantum computers, quantum simulations, atomic clocks and some quantum sensors), thus offering early opportunities for developing and providing QICT services; nevertheless, there are still bottlenecks in quantum technologies.

In quantum communications, for instance, the development of quantum repeaters represents a key technological breakthrough that would allow the exploitation of long-distance QKD and distributed quantum networks. In quantum computing, random fluctuations have to be controlled as they occasionally flip or randomize the state of qubits during processing. Quantum software ecosystems are very active but rather fragmented: efforts are mainly directed to define languages enabling programmers to work with high-level abstractions. Among the other challenges for accelerating this trend, there is a lack of a common terminology and language, which are essential to reach standard solutions. A commonly shared terminology will enable governments, industry, and innovation communities to more effectively interact and operate towards common goals for developing industrial quantum ecosystems.

Moreover, another major obstacle hindering developments and large deployments of QICT infrastructure is the fact that industry choices have not yet consolidated one type of quantum hardware. A quantum hardware abstraction layer (Quantum-HAL) would simplify and speed-up the development of quantum platforms, services, and applications. As in traditional computing systems, a hardware abstraction layer is an abstract layer of programming that allows a computer operating system to interact with a hardware device at an abstract level (rather than at a detailed hardware level); similarly, the Quantum-HAL extends this concept for quantum computing and networking.

Eventually in the cybersecurity areas regarding ICT and telecommunication infrastructures, there is also a growing interest in post-quantum cryptography (PQC). PQC

does not use any quantum technologies, but it refers to cryptographic algorithms that are thought to be secure against a cryptanalytic attack by quantum computers. QKD and PQC have different applicability scenarios in future quantum infrastructures, but their potential end-to-end integration (still under study) is very attractive.

Concerning the most recent priorly stated methods, in [9], four different methods of interconnecting remote QKD networks are proposed. The proposed approaches combine, in a transparent way, different fiber and satellite physical media, as well as common standards of key–delivery interfaces. The testbed interconnections are designed to increase security by utilizing multipath techniques and multiple hybridizations of QKD and PQC algorithms.

## 3. Management and Control of QKD Networks

In context of the Euro-QCI initiative (https://digital-strategy.ec.europa.eu/en/policies/european-quantum-communication-infrastructure-euroqci, accessed on 10 December 2023) (Euro Quantum Communication Infrastructure), the project EQUO is contributing to the realization of a European industrial ecosystem for secure QCI technologies and systems.

EQUO's main focus is the industrialization of European QKD solutions ready for deployment, realizing high-TRLs [8,9] and high-performance technology components like the quantum random number generator and single photon detector as examples. Indeed, system integration from the QKD system to telecommunication networks providing high-level security encryption together with suitable key management represents the key to the application of the whole designed system in real scenarios, assuring interoperability between quantum and traditional cybersecurity systems and protecting sensitive digital data.

Achieving the TRLs [8,9] necessarily requires the design and development of a control and management architecture for the QKD network (QKDN).

EQUO architectural solutions are compliant (by design) to the most relevant international recommendations (ITU-T, Y.38xx "Quantum Key Distribution Networks" series) and European standard (ETSI, Group Specification documents on QKD), to maximize interoperability both between QKDN components (e.g., QKD module, key manager, QKD controller, QKD manager and cryptographic application) and between different QKDNs.

The functional architecture of the QKDN, designed and implemented in a metropolitan use case, is shown in Figure 1. This architectural model is derived from the recommendation ITU-T Y.3800 "Overview on networks supporting quantum key distribution" [10] and subsequent recommendations, ITU-T Y.3801 "Functional requirements for quantum key distribution networks" [11], ITU-T Y.3802 "Quantum key distribution networks—Functional architecture" [12], ITU-T Y.3803 "Quantum key distribution networks—Key management" [13] and ITU-T Y.3804 "Quantum key distribution networks—Control and management" [14].

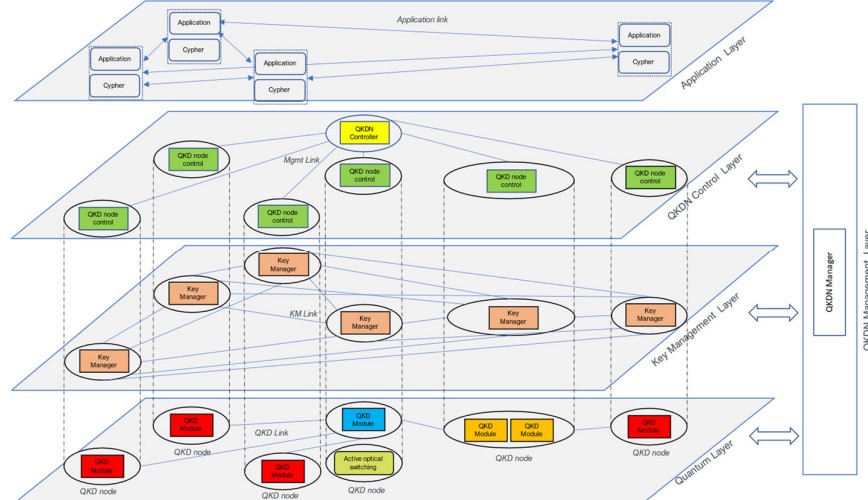

**Figure 1.** EQUO QKDN functional architecture.

The functional architecture also takes into consideration the standards ETSI GS QKD 014 "Quantum Key Distribution (QKD); Protocol and data format of REST-based key delivery API" [15] and ETSI GS QKD 015 "Quantum Key Distribution (QKD); Control Interface for Software Defined Networks" [16].

As extracted from the above-mentioned ITU-T recommendations, a QKDN functional architecture is composed of the following functional layers:

- A quantum Layer—In this layer, each pair of QKD modules connected by a QKD link generates symmetric random bit strings in its own way. Each QKD module pushes the random bit strings up to a key manager (KM) that is in the same QKD node. Each QKD module can also send QKD link parameters (e.g., quantum bit error rates—QBERs, etc.) to the QKDN manager.

- A key management layer—This layer includes KMs and KM links. Each KM is located in a QKD node. The KM performs key management. The KMs are connected via KM links. The KM receives random bit strings from QKD module(s) located in the same QKD node. The KM synchronizes and re-formats these bit strings and stores them as keys in the storage. Interfaces for various cryptographic applications are installed into the KM. The KM receives key requests from a cryptographic application, acquires the necessary number of keys from storage, synchronizes and authenticates the acquired keys via a KM link, and supplies them in an appropriate format to the cryptographic application. If KMs do not have direct KM links between them, they should share the necessary number of keys via key relay. KMs then ask the QKDN controller(s) about an appropriate relay route. Consequently, the keys are transferred and finally supplied to the cryptographic applications. Once the keys have been supplied to the cryptographic applications, the KMs should apply the key management policy, such as by deleting or preserving the keys.

- A QKDN control layer—The QKDN control functions must control QKDN resources to ensure secure and stable operations of a QKDN. QKDN control functions are provided by QKDN Controller(s) and, mainly, include the following:

  - The control of key relay routes including rerouting (e.g., when a failure or eavesdropping occurs) between the two end points of the cryptographic application, which require the key;
  - The control of KMs and KM links;
  - The control of QKD modules and QKD links;
  - Authentication and authorization control;
  - Quality of service (QoS) and charging policy control.

- A QKDN management layer—A QKDN manager located in this layer monitors and manages the QKDN as a whole. The QKDN manager gathers information about the performance of QKD modules and QKD links (including quantum relay points) in the quantum layer and key management information in the key management layer, to monitor these two layers. Moreover, the QKDN manager interacts with the QKDN controller to support the management and control functions of the QKDN. The basic functions are as follows:

  - Fault, accounting, configuration, performance and security (FCAPS) management;
  - The status of whole-QKDN monitoring;
  - Support of key life cycle management in KM;
  - Authentication and authorization management;
  - QoS and charging management.

As previously mentioned, control and management functionalities are essential for the industrialization of QKDN solutions ready for deployment and integration with traditional telecommunication networks. Therefore, EQUO innovation activities focused on these functionalities and identified the relationships between them (the QKDN control layer and QKDN management layer) and the other functional layers.

A feasibility study on the applicability of ITU-T Rec. Y.3804 "Quantum key distribution networks—Control and management" has been pursued as well.

This recommendation specifies control and management functions and procedures for QKDNs, covering the following:

- Functional elements of QKDN control, management, and orchestration;
- Functions of QKDN control, management, and orchestration;
- Procedures of QKDN control, management, and orchestration.

Figure 2 highlights functional components and reference points relevant to QKDN control and management in a QKDN.

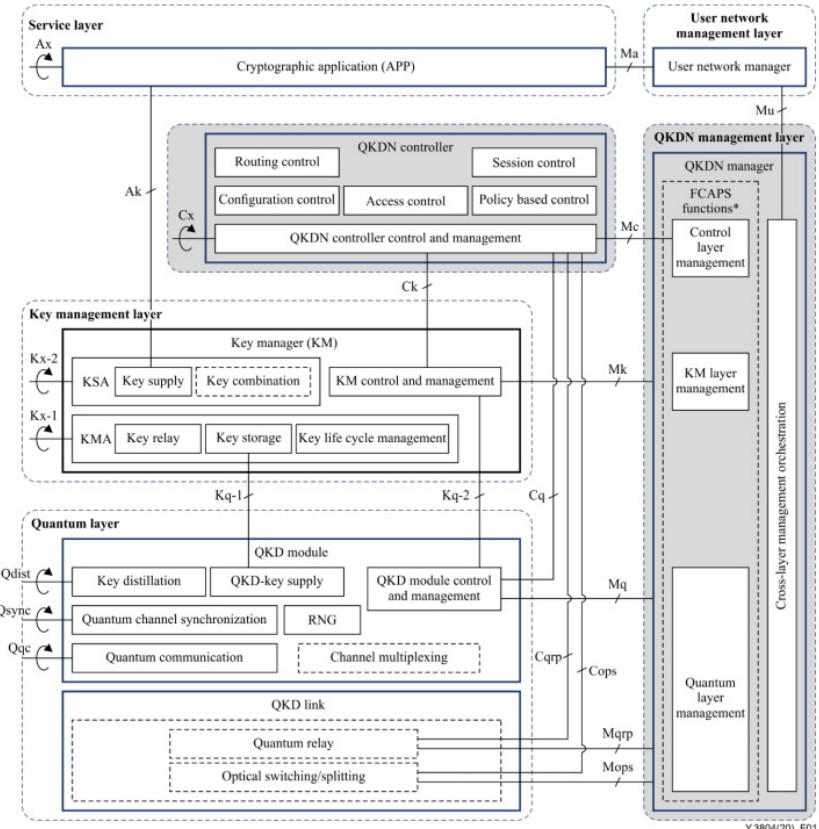

**Figure 2.** Functional elements and reference points relevant to QKDN control and management (source: ITU-T Y.3804). **\***: Traditional fault, configuration, accounting, performance, and security (FCAPS) functionality which is not specific to QKDN is outside the scope of this Recommendation.

The following control functions are supported:

- Routing control—this provisions an appropriate key relay route between two end points of KMs in the key management layer.
- Configuration control—this is responsible for the acquisition of control related configuration information on QKD modules and QKD links in the quantum layer and on KMs and KM Links in the key management layer. Moreover, it controls the state of these components and can reconfigure QKD links and KM links if an alarm or failure diagnosis is noted.
- Access control—this provides capabilities to verify the claimed identity of functional components under the control and support of the QKDN controller and to restrict the functional components to pre-authorized activities or roles;
- Session control—the session control function supports the communication between KMs to establish the end-to-end key relay route or to supply keys to cryptographic

applications in the service layer of the user network. The session control function initiates, maintains, and terminates the session.

- QKDN controller control and management—this is in charge of the overall control and management of the functional elements in the QKDN controller and communicates this with functions in other layers, such as the QKD module, QKD link, KM, and the QKDN manager.

Two scenarios involving the QKDN controller have been investigated. The first one is the provision of a key relay route and the second one is the rerouting of key relay. Both scenarios are compliant with the procedures recommended by ITU-T Y.3804, illustrated in Figures 3 and 4.

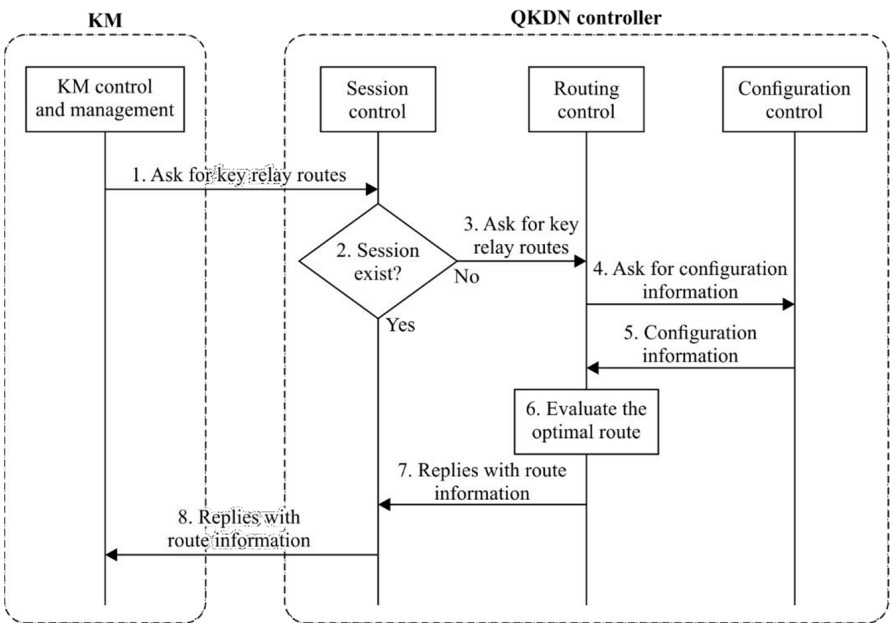

**Figure 3.** Key relay route provisioning procedure (source: ITU-T Y.3804).

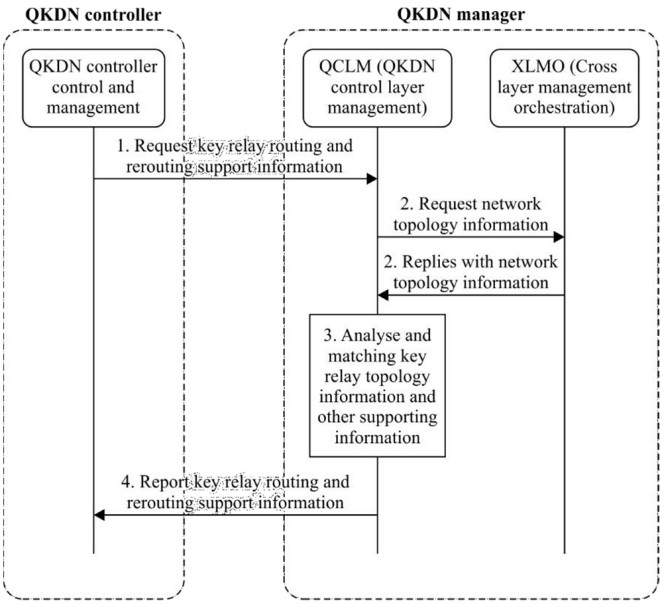

**Figure 4.** Key relay rerouting procedure (source: ITU-T Y.3804).

The QKDN manager supports, for selected scenarios, the following fault management functions:

- Fault monitoring, detection and visualization;
- Fault diagnosis and remedy action definition;
- Support to the QKDN controller for the rerouting control of key relay as needed in the case of faults.

In the example of the use case, two scenarios involving the QKDN manager have been investigated. The first one is QKD link failure (channel loss) and the second one is key relay failure in KM. In compliance with the procedures recommended by ITU-T Y.3804, illustrated by Figures 5 and 6, for both scenarios, the following apply:

- The QKDN manager receives the alarm notification from the faulty resource;
- The QKDN manager applies the rules for identifying the correct healing actions;
- The QKDN manager communicates said actions to the QKDN controller (e.g., key relay path rerouting);
- The QKDN controller actuates said actions.

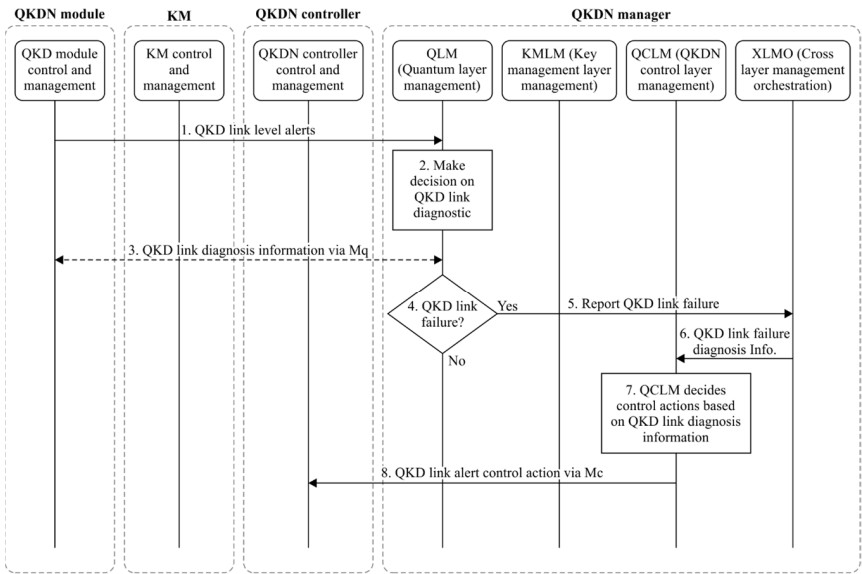

**Figure 5.** QKD link failure procedure (source: ITU-T Y.3804).

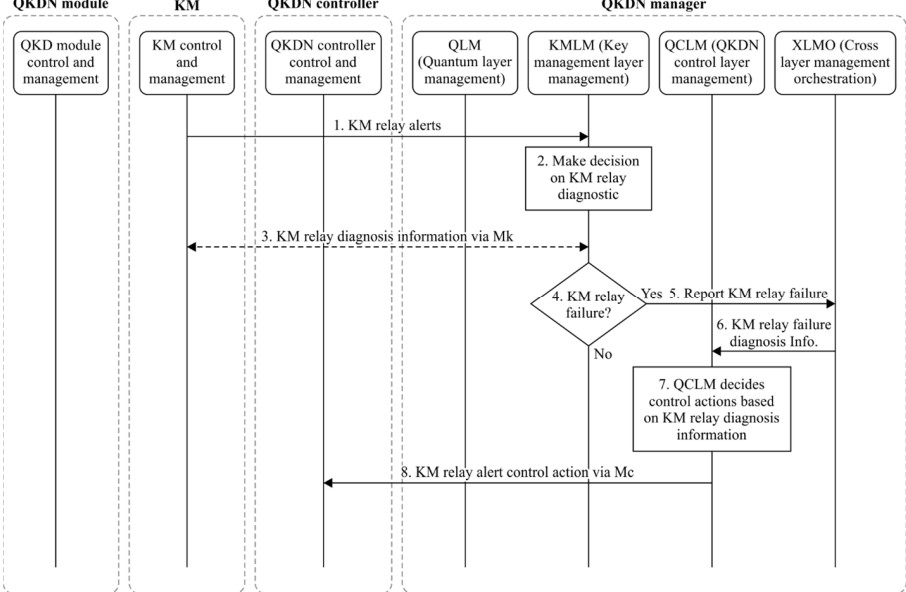

**Figure 6.** Key relay failure in KM procedure (source: ITU-T Y.3804).

The QKDN manager supports, for selected scenarios, the following configuration management functions:

- QKDN resource configuration and provisioning;
- QKDN resource status monitoring;
- QKDN resource inventory management;
- QKDN topology management;
- Support to the QKDN controller for the provision of key relay routes.

Specifically, the QKDN topology discovery scenario has been investigated, in compliance with the procedure recommended by ITU-T Y.3804, illustrated in Figure 7. Each QKDN resource sends to the QKDN manager configuration information that is used to create the QKDN resource inventory and the QKDN topology. For each QKDN resource (e.g., the QKD module, QKD link, and KM), said configuration information contains the following:

- A resource identifier;
- A resource state (e.g., in service, out of service, standby, or reserved);
- Attributes related to the specific characteristic of the resource;
- Relationships with other resources.

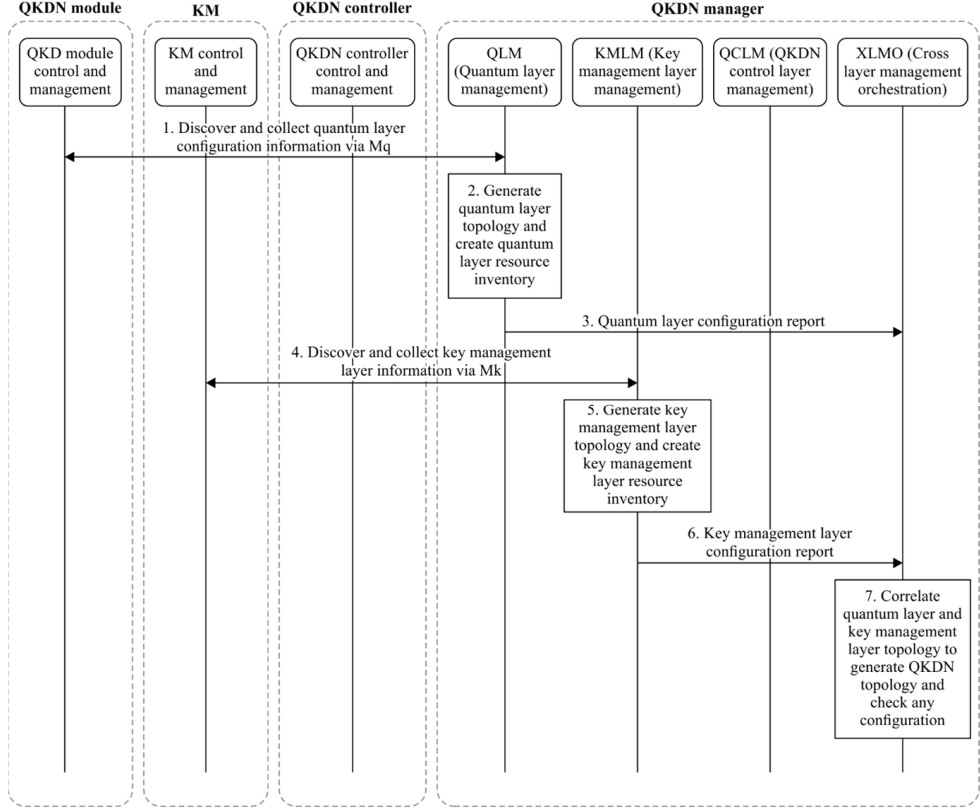

**Figure 7.** QKDN topology discovery procedure (source: ITU-T Y.3804).

The QKDN manager collects all of this information and generates the QKDN topology, considering both the quantum and the key management layers. The QKDN manager keeps the topology updated, gathering every notification coming from the QKDN resources.

The QKDN manager provides updated configuration information to the QKDN controller when required (e.g., in key relay route provisioning and key relay rerouting procedures).

## 4. Next Steps

### 4.1. Hybrid Security (QKD and PQC)

As mentioned in previous sections, QKD is a physical layer method that allows an unconditional secure distribution of random keys between remote users. For instance,

QKD is a method that exploits the physical properties of photons to create and distribute secret keys that can then be used by existing ciphers. In the area of cybersecurity, on the other hand, the goal of PQC is to develop cryptographic systems that are secure against both quantum and classical computers and can interoperate with existing communication protocols and networks. In other words, unlike QKD, PQC relies on algorithms that are too complex for quantum computers to crack. PQC is still in active development, and it is currently undergoing standardization by NIST. Also, the model of the Post Quantum Telco Network is still under definition and development, through several initiatives [17].

There are still limitations in both approaches, and overcoming the current limitations requires a long-term investment in developing QICTs. Nevertheless, it is likely that a combination of QKD and PQC for a so-called hybrid security scenario might be very interesting, offering the most secure approaches to data encryption.

In general, a hybrid mechanism is based on a combination of a recognized pre-quantum public key algorithm and an additional algorithm that is post-quantum-secure. This makes the hybrid mechanism secure thanks to the resistance of the first algorithm against classical attackers and that of the second algorithm against quantum attackers [17]. Specifically, examples of issues under consideration for QKD-PQC hybrid security include understanding which generic standards on hybrid schemes are compatible with QKD-PQC technologies; what kind of QKD-PQC integration can also take place on the authentication level; technical challenges vs. business opportunities. These are for further study.

### 4.2. Quantum Hardware Abstraction Layer (Quantum-HAL)

In the long term, the evolution of QKDN towards the quantum internet will intersect with quantum computing technologies aiming at the functional integration of quantum computing and networking. One major obstacle towards these evolutionary developments and exploitations is that, today, industry choices have not yet consolidated one type of quantum hardware.

In order to mitigate this, a quantum hardware abstraction layer (Quantum-HAL) would allow application and service developers to use the abstractions of the quantum hardware underneath (even if this belongs to different technologies); this would accelerate the development of quantum platforms, services, and applications.

In fact, a Quantum-HAL would provide northbound quantum application programming interfaces for the higher application layers, decoupling from the different types of quantum hardware technologies for quantum computing and networking. This is for further study.

### 4.3. Standardization

There is a need to develop and consolidate a standard solution enabling multi-vendor interworking and interoperability, not only at the physical level but also at the management and control ones. This is mandatory for the mature deployment of QICTs solutions. Multiple groups, such as ANSI, ITU, IETF, ETSI, CEN-CENELEC, GSMA and IEEE, are producing significant synergic efforts. The definition of standard management and control procedures is another key aspect that concerns the integration of future quantum nodes and equipment in classic infrastructure (e.g., optical networks, 5G and beyond); this requires further effort in the definition of abstractions, interfaces and protocols.

## 5. Lessons Learnt

This paper addressed some of the main challenges and opportunities of emerging quantum information and communication Technologies. Specifically, in order to provide a concrete example of innovation activities, the paper focused on quantum-secure communications based on QKD as is being carried out in the European project EQUO.

Some of the main lessons learnt in Euro-QCI and, in general, in other international activities [18–20] include the following:

- Common terminology and language for quantum technologies and services are essential in all steps from innovation to equipment and platform developments and exploitations.
- Technological breakthroughs are required, such as a quantum repeater, the control of errors (error corrections) in quantum computing, and the development of quantum systems supporting entanglement [4].
- There is a need to develop and consolidate a standard architectural solution enabling multi-vendor interworking and interoperability, not only at the physical level but also at the management and control ones. This is mandatory for the mature industrial exploitation of QICTs solutions.
- A quantum hardware abstraction layer may accelerate the development and exploitation of quantum platforms, services, and applications. In fact, today, the industry choices have not yet consolidated one type of quantum hardware.
- QKD and PQC have different applicability scenarios in future quantum infrastructures, but their potential integration in end-to-end cybersecurity services is very promising.

The next steps will address these key challenges in the direction of developing and exploiting the quantum internet [5].

**Author Contributions:** All authors have equally contributed to the paper. All authors have read and agreed to the published version of the manuscript.

**Funding:** The paper has been developed under the project EQUO (European Quantum ecOsystems) funded by the European Commission in the Digital Europe Programme under the grant agreement No. 101091561.

**Data Availability Statement:** Data are contained within the article.

**Conflicts of Interest:** The authors declare no conflict of interest.

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
