# Peer review of "The Rise of Quantum Information and Communication Technologies"

_quantumrep, doi:10.3390/quantum6010003_

Round 1

Reviewer 1 Report

Comments and Suggestions for Authors

The manuscript is a brief summary of recent advances and current challenges in implementing quantum key distribution (QKD) and other quantum information technologies.  The manuscript focuses mainly on control and management of the hardware and on problems that still need to be resolved to make quantum communication technology practical.

The paper is accurate and may serve as a useful guide to the current state of the field.  I would recommend some improvements, however, before it is published.

As a review article, novelty is not expected to be high. However, there should be sufficiently detailed descriptions for a reader from an adjacent field to follow what is being discussed. The manuscript is very telegraphic, mentioning many topics with only a sentence or two of definition and discussion; for example, the section "Hybrid Security" is only two paragraphs long and only mentions hybrid security in the next to last sentence, with no discussion of what it involves, aside from saying it is a "combination of QKD and PQC". (PQC is post-quantum cryptography). More detail here and in many other sections would greatly improve the readability and usefulness of the manuscript. 

I would therefore recommend that the authors add additional detail beyond a cursory definition of each topic discussed, and possibly additional references where more detailed discussions can be found. 

The manuscript is also very acronym-heavy, with (usually multiple) additional acronyms added in every section. Papers in this style are common in engineering journals, but may put off many potential readers of physics journals. Although not mandatory, reducing the number of acronyms may make the paper more user-friendly. 

Finally, the source of Figure 1 should be given (if it is not original to the authors).  

Overall, given the comments above, I would recommend some revision before publication.

Author Response

Dear Reviewer

Thank You very much for reading the paper and for the valuable proposals of revisions. We’ve produced an updated version of the paper fully considering Your proposals.

Specifically, we’ve added more text and references on:

  • State of the art on QKD
  • TRL under section 2
  • Hybrid Security and Post Quantum Telco Network in section 4.1

 We’ve added the following references: [6], [7], [8], [9], [20]

We’ve also reduced the number of acronyms (of about 10), as suggested, to make the paper more user-friendly.

Figure 1 it is original from the Authors of the paper.  

 Please find attached the updated paper with the revisions marks

Thank You

Best Regards

Antonio Manzalini, Luigi Artusio

Reviewer 2 Report

Comments and Suggestions for Authors

In this paper, the authors provide an overview of the EQUO project, describing an architecture for the management of the infrastructure of a quantum key distribution network. In complex, the article is average, but there are some major points that should be improved before publishing the article.

* The main problem of the article is the insufficient introduction and review of the state of the art in the field. The authors do not provide any reference to previous implementations of a quantum key distribution network. A better review on the state of the art on QKDN management is necessary for this work to be publishable.

* There are a lot of missing references in the paper. For example, the authors speak of technology readiness level (TRL) and post-quantum cryptography (PQC) but do not provide any reference to them. Moreover, the authors mention the quantum computing five criteria of Di Vincenzo without referencing any article. The authors should re-check all the article and make sure that they include all the references that could ease the readability of the article.

* Looking at the proposed QKDN scheme, it seems that all the network management is performed in a single management layer. Does it mean that the whole network management is centralized on a single system? How does it compare with the management of classical networks? Maybe it could be interesting to make some further comments on the reasons of the proposed management scheme.

To conclude, the article lacks a proper introduction and state of the art description, with too little references, and therefore it is not suitable for publication. However, once this major issue is addressed, it would represent an interesting description of a proposed QKDN management scheme.

Author Response

Dear Reviewer

Thank You very much for reading the paper and for the valuable proposals of revisions. We’ve produced an updated version of the paper fully considering Your proposals.

We’ve added more text in the introduction. Specifically, a short review on the state of the art of QKD implementation and QKDN management have been introduced with a related reference [9].

Moreover, we’ve added more details also on

  • TRL under section 2
  • Hybrid Security and Post Quantum Telco Network in section 4.1

 In summary we’ve also added the following references: [6], [7], [8], [9] [20]

Some text has been introduces on the questions mentioned in the review report: “the whole network management is centralized on a single system? How does it compare with the management of classical networks? Specifically, it has been mentioned that the approach is based on the Software Defined Networking (SDN) paradigm, where network programmability is enabled through a logically centralized control, management and orchestration planes. This is the trend adopted also for current production telecommunication networks where multiple managers and controllers are deployed for scalability and reliability reasons, which in turn rely on distributed consensus protocols to operate in a logically centralized manner.

Please find attached the updated paper with the revisions marks.

Thank You

Best Regards

Antonio Manzalini, Luigi Artusio

Round 2

Reviewer 1 Report

Comments and Suggestions for Authors

The authors have responded to the previous review by adding a bit more detail to a few of the topics discussed and by making some improvements in the readability. They have added additional references, making it easier for readers to find more detailed treatments of these topics. 

I believe the paper is suitable for publication in its current form.

Reviewer 2 Report

Comments and Suggestions for Authors

In the resubmitted version, the authors have answered in a convincing way to the remarks raised in the previous review and, therefore, the article is publishable in Quantum Reports.